# A Genome-Wide Association Study into the Aetiology of Congenital Solitary Functioning Kidney

**DOI:** 10.3390/biomedicines10123023

**Published:** 2022-11-23

**Authors:** Sander Groen in ’t Woud, Carlo Maj, Kirsten Y. Renkema, Rik Westland, Tessel Galesloot, Iris A. L. M. van Rooij, Sita H. Vermeulen, Wout F. J. Feitz, Nel Roeleveld, Michiel F. Schreuder, Loes F. M. van der Zanden

**Affiliations:** 1Radboud Institute for Health Sciences, Department for Health Evidence, Radboud University Medical Center, 6500 HB Nijmegen, The Netherlands; 2Radboud Institute for Molecular Life Sciences, Department of Paediatric Nephrology, Radboudumc Amalia Children’s Hospital, 6500 HB Nijmegen, The Netherlands; 3Centre for Human Genetics, University of Marburg, 35037 Marburg, Germany; 4Department of Genetics, University Medical Center Utrecht, Utrecht University, 3584 CS Utrecht, The Netherlands; 5Department of Pediatric Nephrology, Emma Children’s Hospital, Amsterdam UMC, University of Amsterdam, 1105AZ Amsterdam, The Netherlands; 6Division of Pediatric Urology, Department of Urology, Radboud Institute for Molecular Life Sciences, Radboudumc Amalia Children’s Hospital, 6500 HB Nijmegen, The Netherlands

**Keywords:** solitary functioning kidney, unilateral kidney agenesis, multicystic dysplastic kidney, kidney hypodysplasia, congenital anomalies of the kidney and urinary tract (CAKUT), genome-wide association study, single nucleotide variant, aetiology, kidney development

## Abstract

Congenital solitary functioning kidney (CSFK) is a birth defect that occurs in 1:1500 children and predisposes them to kidney injury. Its aetiology is likely multifactorial. In addition to known monogenic causes and environmental risk factors, common genetic variation may contribute to susceptibility to CSFK. We performed a genome-wide association study among 452 patients with CSFK and two control groups of 669 healthy children and 5363 unaffected adults. Variants in two loci reached the genome-wide significance threshold of 5 × 10^−8^, and variants in 30 loci reached the suggestive significance threshold of 1 × 10^−5^. Of these, an identified locus with lead single nucleotide variant (SNV) rs140804918 (odds ratio 3.1, *p*-value = 1.4 × 10^−8^) on chromosome 7 was most promising due to its close proximity to *HGF*, a gene known to be involved in kidney development. Based on their known molecular functions, both *KCTD20* and *STK38* could explain the suggestive significant association with lead SNV rs148413365 on chromosome 6. Our findings need replication in an independent cohort of CSFK patients before they can be established definitively. However, our analysis suggests that common variants play a role in CSFK aetiology. Future research could enhance our understanding of the molecular mechanisms involved.

## 1. Introduction

Living with a solitary functioning kidney from childhood predisposes to kidney injury later in life, with up to 80% of children with a solitary functioning kidney showing signs of kidney injury at 18 years of age [1]. Moreover, 20–40% of these children may develop end-stage kidney disease by 30 years of age [2]. Even in less severely affected individuals, signs of kidney injury that develop during childhood can result in a higher risk of cardiovascular disease later in life [3].

A congenital solitary functioning kidney (CSFK) is most often the result of developmental kidney anomalies, such as unilateral kidney agenesis (UKA), multicystic kidney dysplasia (MCDK), or kidney hypo/dysplasia (KHD). These anomalies fall within a spectrum of anomalies referred to as congenital anomalies of the kidney and urinary tract (CAKUT), which are thought to partly have a shared aetiology [4]. Several causative mechanisms may play a role in the aetiology of CAKUT: environmental risk factors, monogenic causes, pathogenic copy number variants (CNVs), and common genetic variants [5].

Environmental risk factors, such as maternal diabetes, overweight, and the use of artificial reproductive technologies, have been shown to play a role in CAKUT aetiology [6,7,8]. For CSFK specifically, our group identified important roles for maternal stress and infections during pregnancy, as well as a protective effect of folic acid supplementation, in addition to previously identified factors such as conception using in vitro fertilization/intracytoplasmic sperm injection, maternal smoking during pregnancy, and older maternal age [9]. Monogenic causes for CAKUT have been discovered in over 150 different genes, and this number keeps rising due to increased availability and decreased costs of exome sequencing [10,11]. Approximately one-third of these genes have been associated with isolated forms of CAKUT, whereas the other genes with defects in multiple organ systems [11]. As CSFK has been reported to be familial in ±10%, and a CSFK may be part of a specific syndrome, several monogenic causes have been found (for a review, please see Westland et al. [12]). Studies performing whole exome sequencing (WES) in patients with sporadic and isolated CSFK are rare, and the yield is relatively low compared to non-isolated or familial CSFK [13]. Wu et al. and Lei et al. identified causative variants in 11% of mostly sporadic and isolated patients with UKA and MCDK, respectively, and Liu et al. solved 7% of patients with isolated UKA or MCDK using a combination of chromosomal microarray analysis and WES [13,14,15]. CNVs also play an important role in the aetiology of CAKUT. This was first identified in a cohort of 192 patients with KHD and later extended to other CAKUT phenotypes, including CSFK [16,17,18]. As CAKUT is between common and rare disorders, low-frequency variants with intermediate effects may explain an important fraction of its genetic aetiology [19]. Nonetheless, genome-wide association studies (GWAS) have rarely been performed for CAKUT because of the large cohorts needed. A GWAS including almost 1400 patients with vesicoureteral reflux (VUR) identified three loci that reached genome-wide significance, with odds ratios ranging from 1.4 to 3.7 [20], whereas a previous GWAS on 500 VUR patients and a GWAS including 756 patients with posterior urethral valves (PUV) failed to identify genome-wide significant loci [21,22]. In contrast, two genome-wide significant variants were identified in a GWAS on whole genome sequencing data from 132 PUV patients: a common variant with a minor allele frequency (MAF) of 0.2 and odds ratio (OR) of 0.4 near *TBX5* and a rare variant with a MAF of 0.05 and OR of 7.2 near *PTK7* [23]. Up to now, no GWAS has been published for CSFK patients. To fill this knowledge gap and investigate the role of common genetic variants in the aetiology of CSFK, we performed a genome-wide association study in a Dutch cohort of patients with CSFK.

## 2. Patients and Methods

### 2.1. Patients

Patients with CSFK were derived from the AGORA (Aetiologic research into Genetic and Occupational/environmental Risk factors for Anomalies in children) data- and biobank [24]. This data- and biobank is coordinated from the Radboud university medical centre and Amalia children’s hospital, and contains clinical information, DNA samples, and questionnaire data from patients with congenital malformations and their parents. Patients were recruited prospectively during a visit to the Department of Paediatric Nephrology or Division of Paediatric Urology of the Amalia children’s hospital or University Medical Centre Utrecht or retrospectively in one of 36 Dutch hospitals participating in the SOFIA (Solitary functioning kidney: aetiology and prognosis) study [1]. For the current GWAS, we selected patients born with a solitary functioning kidney (defined as <20% differential function on MAG-3 or DMSA scans and/or unilateral absence of kidney tissue on kidney ultrasound) resulting from UKA, MCDK or KHD. Patients were selected if they were born between 1 January 1993 and 31 December 2020 and if their DNA was available in the AGORA data- and biobank. Patients with either a known genetic cause or a syndrome (with or without a molecular diagnosis) were excluded from the analyses.

Two healthy control populations were used. In 2011, 39 Dutch municipalities selected random samples of 150 or 300 inhabitants born between 1990–2010 for the AGORA data- and biobank. These children and their parents were invited to participate in AGORA via regular mail. Over 2000 families filled out a questionnaire, while mothers and children from a subsample of 748 families also donated a saliva sample for DNA isolation. Due to this relatively small control population available in the AGORA data- and biobank, we included a second control population of 6468 participants who donated blood samples for the Nijmegen Biomedical Study (NBS) [25]. NBS is a population-based study carried out by the Departments for Health Evidence, Laboratory Medicine, and Human Genetics of the Radboud university medical centre together with the community health service and the municipality of Nijmegen. Participants were recruited via random sampling of inhabitants of Nijmegen who were 18 years or older. Participants filled out one or more questionnaires and donated blood samples. All study protocols were approved by the Regional Committee on Research Involving Human Subjects, and informed consent was obtained from all participants and/or legal representatives.

### 2.2. Genotyping

Genotyping of all patients’ DNA samples was performed by deCODE genetics (Reykjavik, Iceland) using Infinium Global Screening Array deCodeGenetics_V3 (Illumina, San Diego, CA, USA), which contains variants present on the Infinium Global Screening Array (GSA) v3.0 BeadChip as well as 50,000 high-quality markers from the OmniExpress bead chips in order to make data derived from these two chips more compatible. The AGORA controls were genotyped using Infinium Global Screening Arrays, while the NBS controls were genotyped using OmniExpress bead chips (Illumina, San Diego, CA, USA).

### 2.3. Quality Control

Quality control (QC) was performed separately for all three cohorts using the PLINK toolset (https://www.cog-genomics.org/plink/2.0/ accessed on 30 July 2021) [26]. First, a sex check was performed to identify sex discrepancies. Next, variant QC was carried out, removing variants with a call rate <98%, a deviation from Hardy-Weinberg equilibrium with a *p*-value < 1 × 10^−10^ (for patients) or <1 × 10^−6^ (for controls), or a MAF < 0.01. Samples were removed if the call rate was <98%. We used the Kinship-based INference for Gwas (KING) toolset to check family relationships within the cohort and to remove individuals related up to the 3rd degree [27]. 

### 2.4. Imputation

Before imputation, the datasets containing patients and controls were merged. We decided to use a two-stage approach for the analyses. Since the patients and AGORA controls were genotyped using similar chips, we started our analyses with the patients and AGORA controls. Due to a limited number of controls, however, the power to detect an association in this dataset was limited. We had 80% power to detect variants with an allelic OR of 3.2 for a MAF of 0.05, an OR of 2.5 for a MAF of 0.10, and an OR of 2.1 for a MAF of 0.25 with genome-wide significance. In contrast, the NBS cohort had more power given the higher sample size (80% power to detect variants with an OR of 2.9 for MAF of 0.05, OR of 2.3 for MAF 0.10, and OR of 1.9 for MAF 0.25), but could be more prone to array-specific batch effects.

To enable this approach, we created two datasets: the patients were merged with the controls from the AGORA data- and biobank to form the AGORA dataset, and the patients were merged with the controls from the NBS to form the NBS dataset. Imputation was based on the overlap in variants genotyped in patients and controls (*n* = 524,412 in the AGORA cohort and *n* = 207,692 in the NBS cohort). Genotypes of the samples were phased using Eagle (v. 2.4.1) and imputed using Minimac (v. 4) with 1000 Genomes Project Phase 3 data as a reference panel [28,29,30]. After imputation, ancestry estimation was performed using data from 1000 Genomes Project Phase 3 by assigning individuals to closest reference super- and subpopulations according to the geometric distance in the ten principal component space [31]. Further analyses were performed using individuals with predicted GBR or CEU ancestry only to create a homogeneous study population [29].

### 2.5. Statistical Analyses

Using PLINK, ORs were calculated for each SNV using logistic regression models. Analyses were limited to variants with a Minimac imputation quality score (r2) of 0.6 and a minor allele count of 20 (for patients and controls combined). The first four principal components and sex were included as confounders. Inflation was visually inspected using quantile-quantile (QQ) plots, and results were depicted in Manhattan plots, which were created in Rstudio using the ‘qqman’ package. Variants with a *p*-value < 5 × 10^−8^ were considered statistically significant on a genome-wide level, while variants with a *p*-value < 1 × 10^−5^ were considered suggestively significant. FUMA (https://fuma.ctglab.nl/ accessed on 15 April 2022) was used to infer independent genomic risk loci, and we selected loci with at least 5 SNVs showing suggestive significance [32]. To further filter our results and select only the most promising loci, we checked the MAF of the lead SNP in the AGORA and NBS controls and compared these with the MAF in openly available reference sets, such as the Dutch GoNL population [33] or, if GoNL data was not available, data for individuals with European ancestry from the 1000 Genomes Project [29]. We restricted to loci with a lead SNV that: (1) reached genome-wide statistical significance or (2) had a MAF > 0.05 in our cases, and this MAF in our cases was closer to the MAF in the controls of our discovery cohort than to the MAF in the reference database (i.e., using the MAF of the reference database as control would result in a stronger effect of the SNV). For the selected loci, we used LocusZoom to visualise results [34], checked the location (intergenic, intronic, or exonic), and used the Open Target Genetics platform (https://genetics.opentargets.org/ accessed on 19 August 2022) to estimate relatedness with nearby genes. Furthermore, we obtained information on topologically associated domains from TADKB (http://dna.cs.miami.edu/TADKB/ accessed on 12 September 2022) [35], while the RegulomeDB (https://regulomedb.org/ accessed on 12 September 2022) [36] was used to assess the functional importance of loci. Lastly, spatiotemporal and species-specific gene expression was visualised using the web-based application developed by the Kaessmann group (https://apps.kaessmannlab.org/evodevoapp/ accessed on 12 September 2022) [37].

## 3. Results

### 3.1. Participants

The AGORA data- and biobank contained DNA from 184 prospectively collected patients with a solitary functioning kidney and saliva samples from 660 patients collected for the SOFIA study (Figure 1). For 10 participants, both a DNA sample and a saliva sample were retrieved, meaning that 844 samples from 834 unique patients were available for genotyping. Of these 834 patients, 32 were excluded due to failed genotyping, 35 patients were excluded because of a known genetic cause or syndrome and 315 patients presented with a phenotype other than UKA, MCDK, or KHD. This resulted in 452 successfully genotyped CSFK patients. For 669 AGORA controls and 5363 NBS controls, genotyping was performed successfully at an earlier stage.

### 3.2. AGORA Dataset Results

After quality control (see Appendix A) and imputation, the AGORA GWAS dataset consisted of 9,956,431 SNVs in 403 patients and 622 controls. The lambda values (1.03) and visual inspection of the QQplot (Figure 2) showed no signs of inflation. None of the SNVs reached genome-wide significance, but several peaks reached suggestive significance (Figure 3). In total, 43 SNVs had a *p*-value below 1 × 10^−5^, and 11 genomic risk loci were identified. After restricting to loci with at least five SNVs, six were visualized using LocusZoom and listed in Table 1.

### 3.3. NBS Dataset Results

In the NBS dataset, 403 patients and 4366 controls were available for analysis after QC, and 9,313,318 SNVs were present after imputation. Again, no signs of inflation were found based on the lambda value (1.04) and QQplot (Figure 4). Sixteen SNVs from two independent loci reached genome-wide significance (Figure 5). Additionally, 335 SNVs from 42 loci had a *p*-value below 1 × 10^−5^. Twenty-four of these loci contained at least five SNVs and were further investigated (Table 1).

### 3.4. Loci with a Genome-Wide Statistically Significant p-Value

Two loci on chromosomes 7 and 18, reached genome-wide significant *p*-values (Table 1). The locus on chromosome 7, with lead SNV rs140804918, was found to have a MAF of 0.05 in the patients, 0.02 in the NBS controls, and 0.03 in the AGORA controls and the GoNL database, and a corresponding *p*-value of 1.4 × 10^−8^ in the NBS dataset. The resulting ORs in the NBS and AGORA datasets were 3.1 and 1.9, respectively, which are the usual effect sizes for an SNV associated with a congenital malformation [19,38]. Although the SNV is in an intergenic region, the location of this SNV is promising, given its proximity to the Hepatocyte Growth Factor (*HGF)* gene (Figure 6A). Moreover, this genomic risk locus is in the same topologically associated domain as the *HGF* gene, and several of the variants in this locus are located in the regulatory region of *HGF*, with RegulomeDB scores of 0.75–1.0, indicating that they are likely to be regulatory variants [36]. The SNV that we found to be associated could be a marker for such a regulatory variant. *HGF* gene expression in human kidney tissue was found to be highest in weeks 5 and 6 post-conception [37], which aligns with two important events in embryonic kidney development: the appearance of the ureteric buds on day 28 and invasion of the newly developed metanephric mesenchyme on day 32, followed by the start of branching morphogenesis [39].

The lead variant on chromosome 18 was rs184382636, which had allelic odds ratios of 5.9 and 12.9 in AGORA and NBS controls, respectively, and a *p*-value of 1.2 × 10^−10^ in the NBS dataset. The MAF in cases was 0.017, but only 0.002 in the NBS controls and the GoNL database and 0.003 in the AGORA controls, which increases the likelihood of a false positive association. This variant is located in an intergenic region, approximately 500 kbp from the *SMIM21* gene, and has a RegulomeDB score of 0.07 (Figure 6B). *SMIM21* mRNA tissue expression is highest in the testis, with no expression in developing or mature kidney tissue. The SMIM21 protein is thought to be part of the cell membrane [37]. Five other genes are located in the same topologically associated domain, but none could be linked to kidney development. 

### 3.5. Other Selected Loci

Of the thirty other loci (6 from the AGORA dataset and 24 from the NBS dataset, Table 1), we selected the six loci with a lead SNV with a MAF >0.05 in the cases and in which the effect estimate using the reference population MAF would be stronger than the effect estimate identified in the discovery dataset. The MAFs varied between 0.05 and 0.50, and odds ratios ranged between 0.5 and 2.1 (Table 1). Four of the six loci were located in an intergenic region, whereas one was intronic, and one was located just upstream of a gene (Table 2, Figure 6C,D and Figure 7A–D). 

Although two of the intergenic SNVs shared a topologically associated domain with genes involved in kidney development (rs9547854 with *FREM2* and rs148251525 with *VRK1*), these SNVs had low RegulomeDB scores (<0.20, Table 2), suggesting no regulatory effect on the nearby genes. The other two intergenic SNVs could not be linked to genes involved in kidney development. Only two variants were located in or close to protein-coding genes: rs148413365 on chromosome 6 is an intronic variant of *KCTD20*, but also in close proximity to *STK38* (also known as *NDR1*) (Table 2, Figure 6C). Therefore, rs148413365 could be a marker for other variants influencing either of these genes. Both genes show high expression in embryonic kidney tissue and are involved in pathways linked to kidney development. Lastly, rs111283115 on chromosome 15 is a 2kbp upstream variant of *ACAN* (Table 2, Figure 6D). The encoded protein is an important part of the extracellular matrix in cartilage, and causal *ACAN* gene variants have been found in patients with short stature. [41] It has relatively low expression in kidney tissue compared to other genes and could not be linked to kidney development, making it unlikely that variants in this gene are involved in the aetiology of CSFK.

## 4. Discussion

We performed the first GWAS in a phenotypically homogeneous population of patients with CSFK and identified several loci that may be linked to kidney development. Despite the relatively small number of patients included in the GWAS, variants in two loci reached genome-wide statistical significance. In addition, variants in thirty loci reached suggestive significance, of which six were selected as most promising based on the minor allele frequencies seen in our study population and acquired from reference databases. Although we lacked an independent replication cohort, we were able to identify two genomic loci which are most likely to be involved in kidney development based on the result of our GWAS and in silico evaluation.

Of the two loci with genome-wide significance, the locus on chromosome 7 with lead SNV rs140804918 is most interesting because of its MAF, OR, and location close to the *HGF* gene. The involvement of HGF and its receptor Met in kidney development was already suggested by Santos et al. in 1994 [42] and by Woolf et al. in 1995 [43]. Both Hgf and Met proteins were identified in mouse kidneys from embryonic day 11.5 onwards, and branching morphogenesis in cultured kidney cells could be decreased using anti-HGF serum [42,43]. Although complete Hgf or Met knockout is lethal [44], conditional knockout models showed decreased ureteric bud branching and nephron numbers [45]. Indeed, HGF has been identified as the candidate gene in the Munich Wistar Frömter inbred rat that has been shown to have an inherited nephron deficit [46]. Furthermore, Met protein was shown to be involved in branching morphogenesis together with the epidermal growth factor receptor (EGFR) [45], while activation of Met through glial cell line-derived neurotrophic factor (GDNF) was hypothesized to create a positive feedback loop further stimulating branching morphogenesis [33]. Variants affecting HGF protein levels could thus influence branching morphogenesis through activation of Met. 

Only one of the loci with suggestive significance could be linked to kidney development based on in silico analyses. The lead SNV rs148413365 of this locus on chromosome 6 is an intronic variant of the *KCTD20* gene. The molecular function of the encoded KCTD20 protein is not well-characterised but is thought to be within the AKT-mTOR-p70 S6k signalling cascade because of protein-protein-interactions with MAP3K8, PPP1CA, and MARK4 [47,48]. In rodents with reduced ureteric bud branching due to intrauterine growth restriction, AKT-mTOR is one of the pathways that is downregulated [49,50]. In addition, this signalling cascade is crucial for compensatory kidney hypertrophy after a reduction in nephron number [51]. As such, it may also be involved in an adequate response to the maldevelopment of one kidney by the contralateral kidney. In normal human embryonic kidney tissue, activated AKT, mTOR, and P70 S6 kinase are constantly expressed in the ureteric bud [52]. Embryonic tissue from MCDK patients showed different expression patterns in a recent study, indicating that inappropriate mTOR pathway activation may be involved in cyst formation in MCDK [52]. Whether this could be a primary cause of MCDK or a response secondary to, e.g., ureteral blockage is still unclear. The other gene that could possibly explain an association between rs148413365 and CSFK is *STK38.* The STK38 protein was recently identified as part of the Hippo pathway and is able to inhibit YAP/TAZ signalling [53]. Deletion of LATS1/2, which has a similar effect on YAP/TAZ expression as STK38, leads to upregulated YAP/TAZ signalling and kidney agenesis, which could be rescued by reducing YAP/TAZ levels [54]. On the other hand, YAP deletion is known to limit nephrogenesis [55], and YAP and TAZ have been suggested to have dosage-sensitive roles in kidney development [56]. This suggests an important role for YAP/TAZ signalling in kidney development, which could be affected by *STK38* gene variants. Both *KCTD20* and *STK38* are expressed throughout embryonic kidney development and are in the 10% of genes with the strongest expression in foetal kidney tissue [57]. Combined with the evidence of functional relevance, both may be involved in the aetiology of CSFK and should be topics of further investigation. 

An important feature of a GWAS is the unbiased testing of many common variants across the genome. This approach has the advantage that it enables the identification of genes or pathways not previously linked to a certain trait. However, it has the inherent risk of false positive results as well. To reduce this risk, a strict threshold for genome-wide significance (5 × 10^−8^) is generally applied, and replication in an independent cohort is required before an association is established. In our study, two variants reached the genome-wide significance threshold. Unfortunately, we were unable to find an independent cohort with a large enough number of CSFK patients to replicate our findings, which is an important limitation of our study. Although replication in other populations remains needed, the fact that the MAFs of the identified SNVs were consistent across different control populations reduces the chance of type one errors. 

Ideally, our analyses would have been further stratified based on subtypes (*i.e.,* UKA, MCDK, and KHD analysed separately), but this would have led to insufficient power to identify relevant associations. Although patients with known disease-causing mutations or syndromes were excluded, we were unable to investigate the presence of monogenic causes or pathogenic CNVs in most patients (~75%), which would have increased our ability to identify SNVs involved in CSFK aetiology. A strength of our study was the large and homogeneous study population, including only patients with CSFK due to UKA, MCDK or KHD, which are phenotypes likely sharing a common aetiology. Furthermore, we restricted our analyses to participants with the two most common ancestral backgrounds and were able to use two large geographically matched control populations.

Our study provides important knowledge about the role of common variants in the aetiology of CSFK. This may ultimately provide us with new insights into the pathophysiology of unilateral kidney maldevelopment and help explain the CSFK to patients and their parents in clinical practice. The limitations of our study illustrate, however, that establishing large, well-characterized patient cohorts remains important to replicate GWAS findings and identify additional variants. As an example, at least 3000 genotyped patients would be needed for 80% power to identify a variant with an OR of 1.5 and a MAF of 0.05 in the case of a 1:5 case-control ratio. As such, unravelling the aetiology of relatively rare congenital anomalies, such as CSFK, will only be possible if large international collaborations are created. The establishment of the European Reference Networks (e.g., ERKNet and eUROGEN) and the intended foundation of a European Health Data Space will hopefully facilitate such initiatives. 

## 5. Conclusions

In conclusion, we identified several variants that reached or approached genome-wide significance in our GWAS of CSFK. Variants in two loci reached the genome-wide significance threshold of 5 × 10^−8^. Of these, the identified locus with lead SNV rs140804918 on chromosome 7 was most promising due to its close proximity to *HGF*, a gene of which the role in kidney development could be larger than expected so far. In addition, variants in 30 loci reached the suggestive significance threshold of 1 × 10^−5^. Of these, SNV rs148413365 on chromosome 6 was most promising as both *KCTD20* and *STK38* could explain the suggestive significant association based on their molecular functions. Due to the lack of an independent replication cohort, these findings need replication before they can be established, and future research is warranted to enhance our understanding of the molecular mechanisms involved in the origin of CSFK.

## Figures and Tables

**Figure 1 biomedicines-10-03023-f001:**
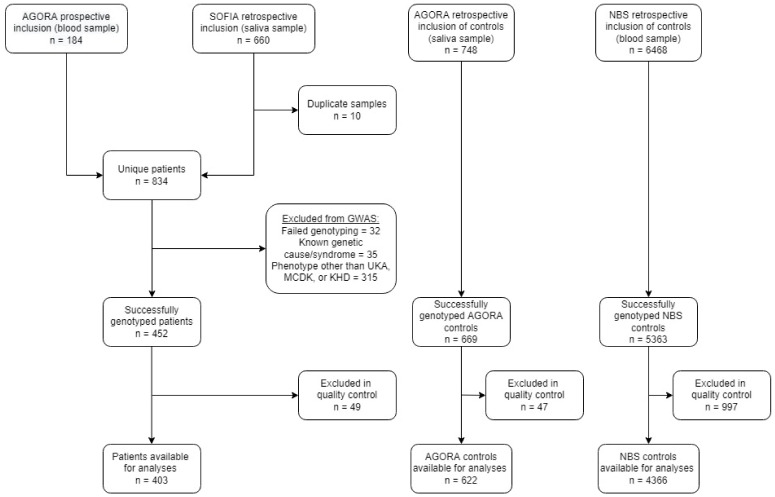
Flowchart of the number of included patients and controls after recruitment, genotyping and quality control. GWAS: genome-wide association study; UKA: unilateral kidney agenesis; MCDK: multicystic dysplastic kidney; KHD: kidney hypo/dysplasia.

**Figure 2 biomedicines-10-03023-f002:**
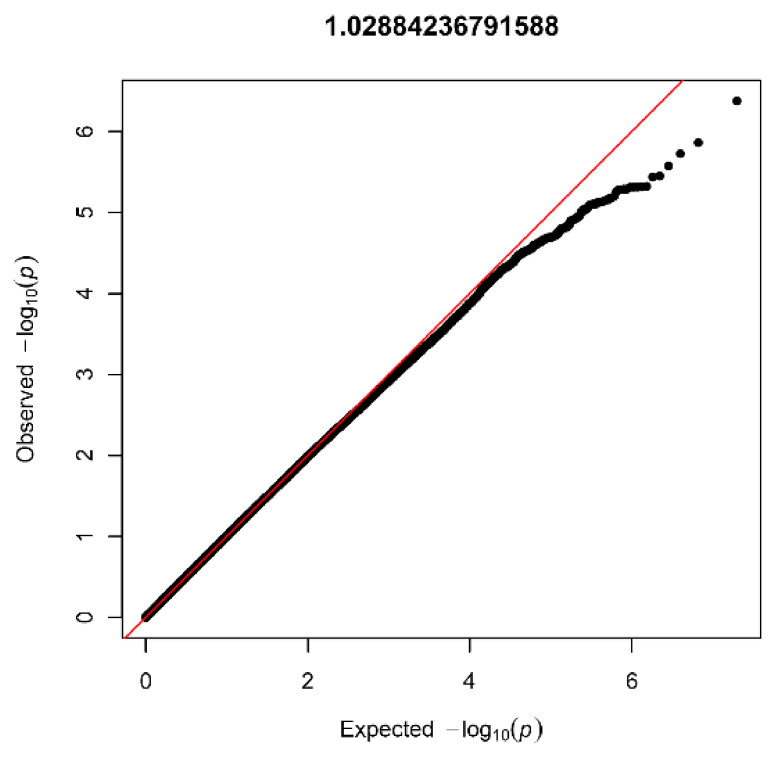
Quantile-quantile plot with lambda value for the AGORA dataset (403 patients and 622 controls).

**Figure 3 biomedicines-10-03023-f003:**
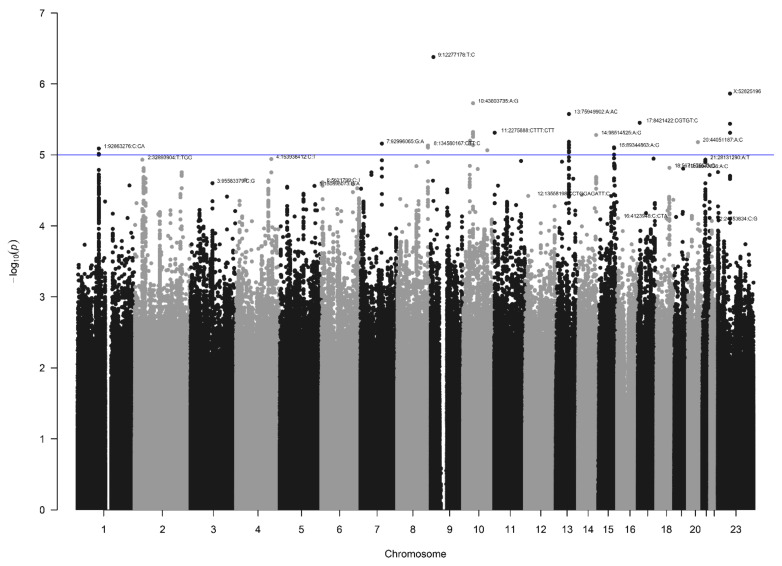
Manhattan plot for the AGORA dataset (403 patients and 622 controls). The blue line indicates the threshold for suggestive statistical significance (*p* < 1 × 10^−5^).

**Figure 4 biomedicines-10-03023-f004:**
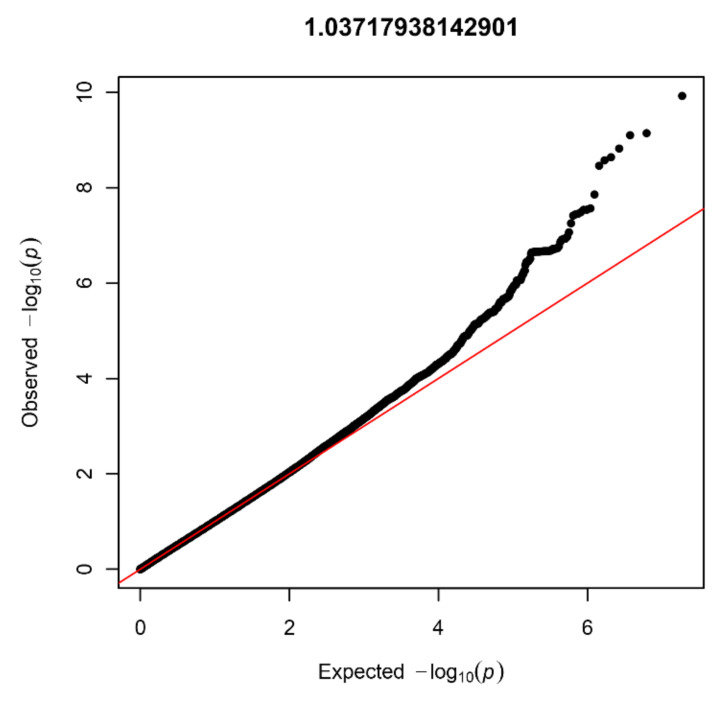
Quantile-quantile plot with lambda value for the NBS dataset (403 patients and 4366 controls).

**Figure 5 biomedicines-10-03023-f005:**
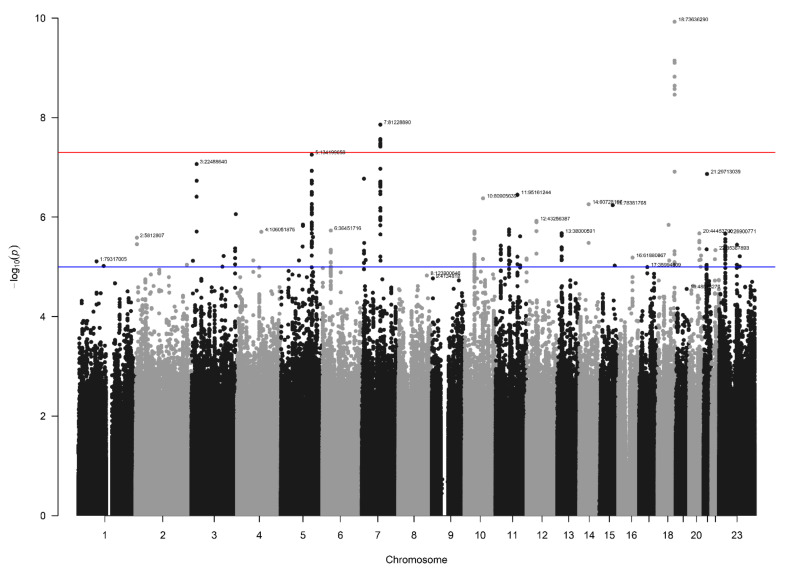
Manhattan plot for the NBS dataset (403 patients and 4366 controls). The blue line indicates the threshold for suggestive statistical significance (*p*-value < 1 × 10^−5^) and the red line indicates the threshold for genome-wide statistical significance (*p*-value < 5 × 10^−8^).

**Figure 6 biomedicines-10-03023-f006:**
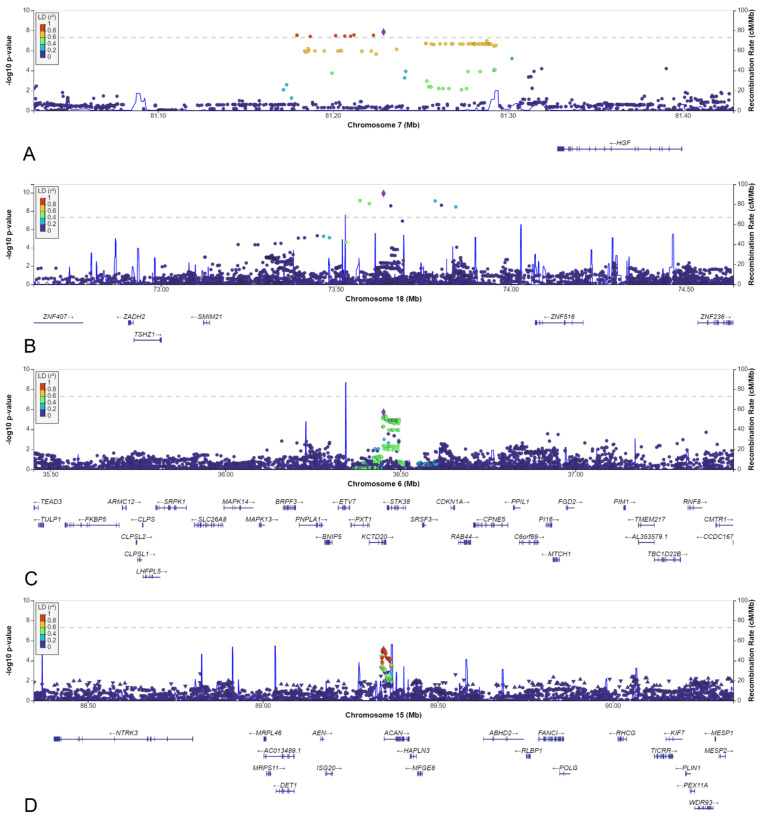
LocusZoom plots of: (**A**) SNV rs140804918 on chromosome 7. (**B**) SNV rs184382636 on chromosome 18. (**C**) SNV rs148413365 on chromosome 6. (**D**) SNV rs111283115 on chromosome 15.

**Figure 7 biomedicines-10-03023-f007:**
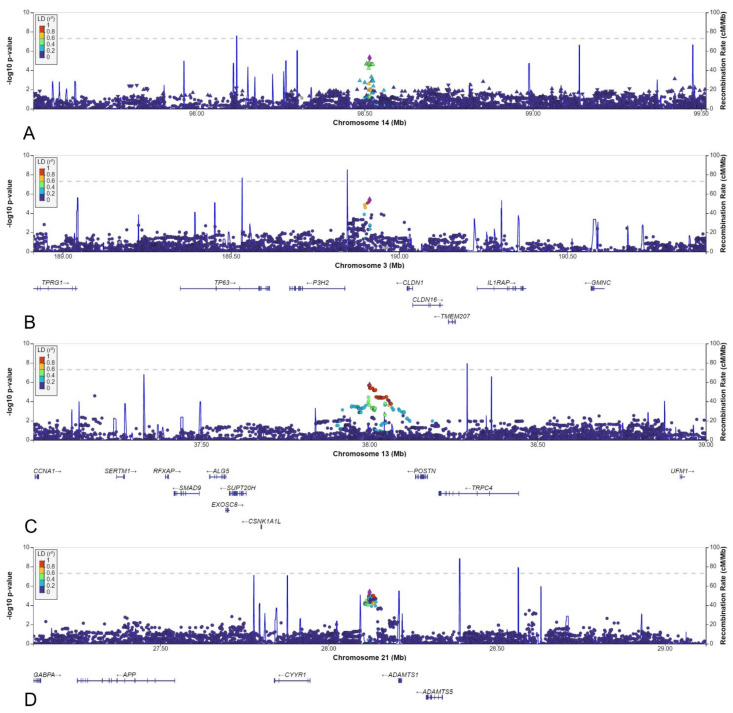
LocusZoom plots for: (**A**) rs148251525. (**B**) rs10433490. (**C**) rs9547854. (**D**) rs2830456.

**Table 1 biomedicines-10-03023-t001:** Genomic risk loci with at least five SNVs with a *p*-value below 1 × 10^−5^ identified in a genome-wide association study of congenital solitary functioning kidney, with the lead SNV for each locus being shown.

Chr	Position	rsID	Discovery Dataset	MAF Patients	MAF Reference	AGORA Dataset			NBS Dataset			n SNVs
						MAF Controls	OR	*p*-Value	MAF Controls	OR	*p*-Value	
Genome-wide significant SNVs												
7	81228890	rs140804918	NBS	0.048	0.031	0.029	1.91	8.4 × 10^−3^	0.020	3.10	1.4 × 10^−8^	51
18	73636290	rs184382636	NBS	0.017	0.002	0.003	5.93	2.4 × 10^−3^	0.002	12.9	1.2 × 10^−10^	42
**SNVs selected based on MAF in discovery and reference populations**												
3	189912222	rs10433490	NBS	0.092	0.169	0.153	0.59	4.0 × 10^−4^	0.150	0.53	4.3 × 10^−6^	6
6	36451716	rs148413365	NBS	0.106	0.188 *	0.156	0.94	7.0 × 10^−1^	0.166	0.51	1.9 × 10^−6^	84
13	38000591	rs9547854	NBS	0.087	0.047	0.052	1.83	1.6 × 10^−3^	0.051	1.97	2.1 × 10^−6^	66
14	98514525	rs148251525	AGORA	0.127	0.048	0.075	2.08	5.2 × 10^−6^	0.106	1.22	8.2 × 10^−2^	5
15	89344863	rs111283115	AGORA	0.333	0.449	0.499	0.65	7.8 × 10^−6^	0.387	0.71	3.5 × 10^−4^	24
21	28122146	rs2830456	NBS	0.281	0.370	0.410	0.64	2.5 × 10^−5^	0.350	0.63	4.5 × 10^−6^	46
**Other suggestively significant SNVs**												
1	79317005	rs140198115	NBS	0.025	0.009	0.011	2.37	1.9 × 10^−2^	0.010	3.79	7.8 × 10^−6^	17
1	92663276	rs11406716	AGORA	0.319	0.283 *	0.246	1.64	8.1 × 10^−6^	0.277	1.27	8.4 × 10^−3^	76
3	22488640	rs182382581	NBS	0.025	0.012	0.009	4.10	1.5 × 10^−3^	0.008	5.04	8.6 × 10^−8^	7
3	6328767	rs7629003	NBS	0.009	0.003	0.003	4.68	4.9 × 10^−2^	0.002	14.48	7.6 × 10^−6^	20
4	106051876	rs190397903	NBS	0.020	0.007	0.013	1.97	8.5 × 10^−2^	0.007	5.23	2.0 × 10^−6^	9
4	70836149	rs146356251	NBS	0.018	0.008	0.008	2.79	2.1 × 10^−2^	0.006	4.84	7.4 × 10^−6^	7
5	78674386	rs138487999	NBS	0.034	0.019	0.016	2.33	5.5 × 10^−3^	0.014	2.84	7.4 × 10^−6^	125
5	139646076	rs145922969	NBS	0.030	0.025	0.013	3.15	1.6 × 10^−3^	0.015	3.20	2.5 × 10^−6^	8
5	96015902	rs181945740	NBS	0.018	0.001	0.008	4.20	7.0 × 10^−3^	0.005	5.90	1.4 × 10^−6^	7
5	134199058	rs73282857	NBS	0.019	0.006	0.005	4.68	9.4 × 10^−3^	0.006	7.00	5.6 × 10^−8^	150
7	10014656	rs190937828	NBS	0.090	0.063	0.066	1.76	1.3 × 10^−3^	0.054	2.30	1.7 × 10^−6^	14
8	134595232	rs10112722	AGORA	0.437	0.366	0.347	1.55	7.5 × 10^−6^	0.370	1.34	2.3 × 10^−4^	13
10	44117049	rs60001082	NBS	0.085	0.069 *	0.065	1.46	4.3 × 10^−2^	0.054	2.00	7.3 × 10^−6^	17
11	2275888	rs1325019515	AGORA	0.380	n/a	0.305	1.61	4.9 × 10^−6^	0.312	1.40	1.1 × 10^−4^	19
11	58767505	rs11229723	NBS	0.022	0.006	0.011	2.23	3.4 × 10^−2^	0.007	4.41	1.8 × 10^−6^	64
11	95161244	rs192152837	NBS	0.018	0.004	0.004	10.2	2.4 × 10^−4^	0.005	6.35	3.6 × 10^−7^	8
12	43256387	rs181072352	NBS	0.012	0.005	0.008	1.44	4.8 × 10^−1^	0.002	10.40	1.2 × 10^−6^	5
12	385846	rs554290784	NBS	0.015	0.005	0.004	4.23	1.1 × 10^−2^	0.003	5.59	6.8 × 10^−6^	5
13	75949902	rs144419778	AGORA	0.267	0.212 *	0.19	1.71	2.7 × 10^−6^	0.205	1.47	1.9 × 10^−5^	24
14	60728107	rs146557102	NBS	0.041	n/a	0.018	2.75	4.8 × 10^−4^	0.017	2.96	5.5 × 10^−7^	8
18	50196707	rs77493404	NBS	0.022	0.009	0.014	2.25	1.2 × 10^−2^	0.008	4.23	7.5 × 10^−6^	22
20	44453790	rs201841698	NBS	0.033	0.029	0.020	2.42	6.1 × 10^−3^	0.013	3.40	2.1 × 10^−6^	7
20	51885312	rs6126804	NBS	0.030	0.008	0.016	2.58	7.3 × 10^−3^	0.013	3.30	6.0 × 10^−6^	8
22	32945612	rs73172143	NBS	0.095	0.078	0.068	1.47	2.7 × 10^−2^	0.059	1.88	7.0 × 10^−6^	6

* MAF from European reference population due to absence in the GoNL database. Chr: chromosome; rsID: Reference SNV cluster ID; MAF: minor allele frequency; OR: odds ratio; n SNVs: amount of single nucleotide variants in associated genomic risk locus; n/a: not available.

**Table 2 biomedicines-10-03023-t002:** A selection of genomic risk loci with candidate affected protein-coding genes.

Chr	rsID	Dataset	Candidate Affected Gene	Distance from SNV to TSS	Regulome Score [36]	Variant Effect Predictor [40]	Gene Function *
Genome-wide significant SNVs							
7	rs140804918	NBS	*HGF*	171 kbp	0.247	Intergenic	Regulates cell growth, motility and morphogenesis in various cell types and tissues
18	rs184382636	NBS	*SMIM21*	497 kbp	0.071	Intergenic	Integral membrane component
SNVs selected based on MAF in discovery and reference populations							
3	rs10433490	NBS	*P3H2*	72 kbp	0.154	Intergenic	Posttranslational modifier of collagen IV
6	rs148413365	NBS	*KCTD20*	41 kbp	0.163	Intronic	Member of AKT-mTOR-p70 S6k signalling cascade
			*STK38*	64 kbp			YAP/TAZ inhibitor in hippo pathway
13	rs9547854	NBS	*POSTN*	172 kbp	0.112	Intergenic	Induces cell attachment and spreading and plays a role in cell adhesion
14	rs148251525	AGORA	*BCL11B*	1224 kbp	0.000	Intergenic	T cell transcription factor
15	rs111283115	AGORA	*ACAN*	1.8 kbp	0.145	Upstream gene variant	Major component of extracellular matrix of cartilaginous tissues
21	rs2830456	NBS	*ADAMTS1*	96 kbp	0.184	Intergenic	Has antiangiogenic activity, involved in inflammation and cancer cachexia

* Based on https://www.ncbi.nlm.nih.gov/gene/. Chr: chromosome; rsID: Reference SNV cluster ID; SNV: single nucleotide variant; TSS: transcription start site; kbp: kilo base pair; MAF: minor allele frequency.

## Data Availability

The data presented in this study are available on request from the corresponding author. The data are not publicly available to protect the privacy of the study participants.

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
