# Peer review of "A Genome-Wide Association Study into the Aetiology of Congenital Solitary Functioning Kidney"

_biomedicines, 2022, doi:10.3390/biomedicines10123023_

Round 1

Reviewer 1 Report

The authors performed a GWAS searching for genetic polymorphisms associated with congenital solitary functioning kidney.

Introduction- the genetic basis of congenital solitary functioning kidney should be described in more detail.

Inclusion and exclusion criteria should be described more precisely.

The information about the functional role of detected polymorphisms is needed.

The authors should discuss the potential clinical significance of the results obtained.

Author Response

We would like to thank the Reviewers for their suggestions to improve our manuscript. Please find a point-by-point response below.

The authors performed a GWAS searching for genetic polymorphisms associated with congenital solitary functioning kidney.

  1. Introduction- the genetic basis of congenital solitary functioning kidney should be described in more detail.

Thank you for the suggestion to include more information on the genetic basis of congenital solitary functioning kidney. In order to make sure that the size of the Introduction was still acceptable, we added a sentence and referred to a previous review with more detail on the genetics of CSFK: “As CSFK has been reported to be familial in ±10%, and a CSFK may be part of a specific syndrome, several monogenic causes have been found (for a review, please see Westland et al.).” (page 3, last paragraph)

  1. Inclusion and exclusion criteria should be described more precisely.

We agree that in- and exclusion criteria were not very clear and scattered in the first paragraph om the methods section. We now clustered all information about in- and exclusion criteria: “For the current GWAS, we selected patients born with a solitary functioning kidney (defined as <20% differential function on MAG-3 or DMSA scans or unilateral absence of kidney tissue on kidney ultrasound) resulting from UKA, MCDK or KHD. Patients were selected if they were born between January 1st 1993 and December 31st 2020 and if their DNA was available in the AGORA data- and biobank. Patients with either a known genetic cause or a syndrome (with or without molecular diagnosis) were excluded from the analyses.” (page 5, last 5 sentences of first paragraph).

  1. The information about the functional role of detected polymorphisms is needed.

We added to the results section (page 10, first paragraph) that the statistically significant SNV on chromosome 18 is located in intergenic region. In addition, we added: “The SNV that we found to be associated could be a marker for such a regulatory variant.”

For rs148413365, we added “Therefore, rs148413365 could be a marker for other variants influencing either of these genes.” To the results section (page 11, second paragraph).

  1. The authors should discuss the potential clinical significance of the results obtained.

Results from genome wide association studies may provide an insight into the pathophysiology of a disease/condition. With the lack of a replication cohort, we need to be careful not to overstate our findings. All in all, we feel that the relevance for daily clinical practice is not reached yet. In order to reflect this, we have added in our Discussion: “This may ultimately provide us with new insights into the pathophysiology of unilateral kidney maldevelopment, and help in providing an explanation for the CSFK to patients and their parents in clinical practice.” (page 14, 2nd paragraph)

Reviewer 2 Report

This is an interesting report with well writing. I believe this manuscript is suitable to be published. However, I have only few comments and questions. Please move the Supplementary Material Figure S1 to S4 to the main text. Please combine Figure 6~8 to one figure, and so did Figure S1~S4. It is required to cite more references to support the authors' findings in discussion section. I will recommend to provide the independent conclusion section (Line 337-344 to be as "5. conclusion"), and please add few more key findings in conclusion. 

Author Response

We would like to thank the Reviewers for their suggestions to improve our manuscript. Please find a point-by-point response below.

This is an interesting report with well writing. I believe this manuscript is suitable to be published. However, I have only few comments and questions.

  1. Please move the Supplementary Material Figure S1 to S4 to the main text.

We moved the figures to the main text (figure 7A-D).

  1. Please combine Figure 6~8 to one figure, and so did Figure S1~S4.

We combined figure 6-9 to figure 6A-6D and figure S1-S4 to figure 7A-7D.

  1. It is required to cite more references to support the authors' findings in discussion section.

Thank you for the suggestion to include more references. We have added several throughout the Discussion.

  1. I will recommend to provide the independent conclusion section (Line 337-344 to be as "5. conclusion"), and please add few more key findings in conclusion.

We made a separate “conclusion” section (page 15) and added: “Variants in two loci reached the genome-wide significance threshold of 5x 10-8 . Of these, the identified locus with lead SNV rs140804918 on chromosome 7 was most promising due to its close proximity to HGF, a gene of which the role in kidney development could be larger than expected so far. In addition, variants in 30 loci reached the suggestive significance threshold of 1x 10-5. Of these, SNV rs148413365 on chromosome 6 was most promising as both KCTD20 and STK38 could explain the suggestive significant association based on their molecular functions”

Round 2

Reviewer 2 Report

This manuscript is ready to publish.